# Carotid Plaque Vulnerability Diagnosis by CTA versus MRA: A Systematic Review

**DOI:** 10.3390/diagnostics13040646

**Published:** 2023-02-09

**Authors:** Konstantinos Dakis, Petroula Nana, Chaidoulis Athanasios, Konstantinos Spanos, Batzalexis Konstantinos, Athanasios Giannoukas, George Kouvelos

**Affiliations:** Department of Vascular Surgery, University Hospital of Larissa, Faculty of Medicine, School of Health Sciences, University of Thessaly, 41110 Larissa, Greece

**Keywords:** carotid, plaque, intraplaque hemorrhage, plaque ulcer, thin-fibrous cap, MRA, CTA

## Abstract

Stenosis grade of the carotid arteries has been the primary indicator for risk stratification and surgical treatment of carotid artery disease. Certain characteristics of the carotid plaque render it vulnerable and have been associated with increased plaque rupture rates. Computed tomography angiography (CTA) and magnetic resonance angiography (MRA) have been shown to detect these characteristics to a different degree. The aim of the current study was to report on the detection of vulnerable carotid plaque characteristics by CTA and MRA and their possible association. A systematic review of the medical literature was executed, utilizing PubMed, SCOPUS and CENTRAL databases, according to the Preferred Reporting Items for Systematic Reviews and Meta-Analysis (PRISMA) 2020 guidelines. The study protocol has been registered to PROSPERO (CRD42022381801). Comparative studies reporting on both CTA and MRA carotid artery studies were included in the analysis. The QUADAS tools were used for risk of bias diagnostic imaging studies. Outcomes included carotid plaque vulnerability characteristics described in CTA and MRA and their association. Five studies, incorporating 377 patients and 695 carotid plaques, were included. Four studies reported on symptomatic status (326 patients, 92.9%). MRA characteristics included intraplaque hemorrhage, plaque ulceration, type VI AHA plaque hallmarks and intra-plaque high-intensity signal. Intraplaque hemorrhage detected in MRA was the most described characteristic and was associated with increased plaque density, increased lumen stenosis, plaque ulceration and increased soft-plaque and hard-plaque thickness. Certain characteristics of vulnerable carotid plaques can be detected in carotid artery CTA imaging studies. Nevertheless, MRA continues to provide more detailed and thorough imaging. Both imaging modalities can be applied for comprehensive carotid artery work-up, each one complementing the other.

## 1. Introduction

Εxtracranial carotid artery disease is historically reported to be responsible for approximately 15–20% of acute ischemic strokes, amaurosis fugax and transient ischemic attacks (TIA) [1]. High-grade stenosis of the extracranial carotid artery has been considered the main parameter justifying surgical revascularization, with either carotid endarterectomy (CEA) or carotid artery stenting (CAS). As of 2015, a total of 34 guideline statements from numerous cardiovascular societies have endorsed lumen stenosis, while synchronously incorporating symptomatic status and surgical risk, as the main factors guiding surgical intervention [2].

However, moderate (>50%) carotid artery lumen stenosis is associated with an annual stroke risk of <1%, thus providing grounds under which lumen stenosis alone is not an absolutely reliable surveillance and stratification tool [3,4]. Intraplaque hemorrhage (IPH), plaque ulceration, plaque neovascularity, thin fibrous cap (FC) and lipid-rich necrotic core (LRNC) have been considered as “vulnerable” plaque characteristics and they have been related to increased rates of plaque rupture and cerebrovascular microembolization [5,6]. 

Subsequently, imaging modalities applied towards carotid artery disease evaluation, including duplex ultrasound (DUS), computed tomography angiography (CTA) and magnetic resonance angiography (MRA), of the carotid arteries have been focusing in detecting these characteristics, in addition to the degree of lumen stenosis [7,8,9]. The benefits and drawbacks of each imaging modality, regarding their diagnostic accuracy in detecting these characteristics, have not been comparatively assessed to produce robust results regarding the superiority of one method over the other. Carotid artery CTA provides a swift, detailed imaging of the extracranial and intracranial carotid artery systems, while contrast media administration aids in high accuracy detection of atherosclerotic plaques. However, specific MRA investigation protocols provide more detailed imaging due to improved spatial resolution, less saturation effects and intravoxel dephasing and better evaluation of vessel lumen and plaque characteristics. While sonographic evaluation of the carotid arteries is important in cerebrovascular events diagnosis and management, h, CTA and MRA are often fundamental in cases requiring surgical intervention.

The aim of the current review was to systematically search and evaluate the available studies comparing the accuracy of CTA and MRA in diagnosing vulnerable carotid plaque characteristics.

## 2. Methods

### 2.1. Review Protocol

The study protocol for the current review has been registered to PROSPERO (CRD42022381801). The PRISMA (Preferred Reporting Items for Systematic Reviews and Meta-analyses) 2020 Guidelines for Systematic Reviews and Meta-analyses were followed [10]. Randomized and non-randomized comparative observational studies, published up to October 2022, reporting on symptomatic and/or asymptomatic carotid plaque vulnerability characteristics, using both CTA and MRA imaging, were considered eligible. Studies reporting solely on one of the two imaging modalities were excluded from the analysis. Case reports, case series, and studies reporting on experimental diagnostic models or not reporting on humans were not considered eligible. Language was not an exclusion criterion, given that an English copy of the article was made available. Scientific Council approval in terms of ethical considerations was not required due to the nature of the study. Data extraction and methodological assessment was executed by two investigators (K.D. and P.N.). Any disagreement was resolved after a discussion with a third investigator (G.K.). A full-text review of the eligible studies included was conducted, respecting the inclusion and exclusion criteria, accordingly.

### 2.2. Search Strategy

A data search of the medical literature was executed, with an endpoint set for the 28 October 2022. The scientific databases PubMed, SCOPUS and CENTRAL were searched following clinically based questions defined before the initiation of the review under the P.I.C.O. (patient; intervention; comparison; outcome) model [11] (Appendix A). The terms “stroke”, “transient ischemic attack”, “TIA”, “amaurosis fugax”, “computed tomography angiography”, “CTA”, “magnetic resonance angiography”, “MRA”, “intraplaque hemorrhage”, “IPH”, “lipid-rich necrotic core”, “LRNC”, “neovascularization”, “inflammation”, “ulceration”, “thrombus”, “thin-fibrous cap”, “TFC” and “carotid plaque vulnerability” were utilized alone and under the Expanded Medical Subject Headings (MeSH) filter in various combinations (Appendix A). Duplication screening was executed by automation tools (EndNote 20.2.1) as well as by the two independent investigators (K.D., P.N.). Primary selection was constructed on title and abstracts, while a secondary investigation was executed based on a full-text review. 

### 2.3. Data Extraction

A standard Microsoft Excel spreadsheet extraction file was developed. Data extraction included general information (article author, title, year of publication, journal of publication, study type, country of origin, study aim, lesion definition and researcher experience on carotid artery lesion diagnosis). Additionally, clinical information was collected, including the patient number, age, male-to-female ratio, symptomatic of asymptomatic status, symptom type (stroke, TIA and amaurosis fugax), number of plaques evaluated, type of imaging modalities applied, time interval between imaging studies performed, MRA and CTA technical characteristics, sensitivity, specificity, positive prognostic value and negative prognostic value for diagnosis of vulnerable carotid plaque characteristics, applying either one of the included imaging modalities, wherever available. Carotid plaque vulnerability characteristics included the stenosis rate, intraplaque hemorrhage (IPH), mean plaque density, mean soft-plaque density, mean hard-plaque density and mean plaque density. In cases where a statistic significance was observed, it was reported accordingly.

### 2.4. Quality Assessment

The risk of bias of the included studies in the final analysis was executed through the application of the Quality Assessment of Diagnostic Accuracy Studies (QUADAS-2 and QUADAS-C) tools, which are primarily used for quality assessment and applicability in systematic reviews and meta-analyses regarding the accuracy of diagnostic studies [12]. The tools consist of four key domains, including patient selection, index testing, reference standard as well as flow and timing and is applied in four phases. Following application of the tools in each study, a “low”, “high” or “unclear” risk of bias is produced. Quality assessment was carried out by two independent investigators (K.D. and P.N.). In case of disagreement, a third author was advised (G.K.). 

### 2.5. Definitions

Carotid artery stenosis severity was carried out in accordance with the NASCET criteria in all included studies [13]. The American Heart Association classification of atherosclerotic plaque was used in one of the included studies [14]. 

### 2.6. Statistical Analysis

The heterogeneity in outcome report did not allow for a quantitative analysis of data. Thus, only a descriptive review of the data was executed.

## 3. Results

The initial search produced 3302 studies. Duplication screening excluded 174 studies (135 by automatic duplicate exclusion, 39 by manual duplicate exclusion). Title and abstract as well as data overlap screening excluded 46 studies. Finally, full-text scrutiny excluded 46 studies, providing 5 eligible studies for data extraction and analysis (2010–2016) [8,9,15,16,17] (Figure 1).

The included studies incorporated 377 patients and 695 carotid plaques. Four studies provided data regarding patient age (median age: 70.1 ± 9.2 years) and sex (357 patients, 70% males) [8,9,16,17]. Exclusion criteria reported in the studies involved previous carotid artery surgical interventions (carotid endarterectomy and carotid artery stenting), complete carotid artery occlusion, any major contraindication for CTA or MRA (established contrast-median anaphylactic reactions) and low-quality produced imaging studies. Study characteristics are presented in Table 1. Four studies incorporated data regarding symptomatic status of carotid plaque (326 patients, 92.9% symptomatic) [8,9,15,17]. Only one study referred to the presence of stroke or TIA (34 symptomatic patients, 88.2% stroke) [8]. No study reported on the severity of cerebrovascular events (Table 2).

All five studies included patients who underwent both CTA and MRA of the carotid arteries, while one study included patients who underwent digital subtraction angiography (DSA) as the referral imaging modality, as well as both CTA and MRA. MRA, CTA and DSA were utilized as the control imaging modality in three, one and one study, respectively. All studies incorporated blinded evaluation of the produced imaging studies and fully disclosed the number and experience of the researchers who evaluated the imaging studies. The time intervals between the applied imaging modalities and data regarding the technical aspects of both MRA and CTA, as well as the use of contrast median, were reported (Table 3).

In terms of the stenosis evaluation, all studies reported their outcomes based on the NASCET criteria using CTA measurements [8,9,15,16,17]. Four studies reported on the degree of stenosis, incorporating 585 carotid plaques (mild stenosis: 56.5%, moderate stenosis: 20%, severe stenosis: 23.5%) [8,9,15,17].

### 3.1. Carotid Plaque Vulnerability Characteristics

Regarding carotid plaque vulnerability, intraplaque hemorrhage (IPH) was the most evaluated characteristic, which was assessed with MRA in four studies (150 plaques) [8,9,16,17]. U-Kind-Im et al. associated IPH with the severity of stenosis, mean plaque density and plaque ulceration that was displayed in CTA findings. Specifically, IPH-positive plaques were characterized by higher mean plaque density [47HU, SD: 15 vs. 43HU, SD: 14 (*p* = 0.001)] and higher mean NASCET stenosis percentage [58%, SD: 26 vs. 20%, SD: 26 (*p* = 0.02)]. Additionally, plaque ulceration was observed in 45 (80.3%) and 51 (91%) cases of IPH-positive plaques by the two independent readers evaluating the imaging studies [9].

Anzidei et al. compared MRA and CTA findings with DSA as the referral imaging modality. They incorporated two different types of MRA protocols (FP; first-pass, SS; steady state) and compared the diagnostic accuracy of the two imaging modalities based on DSA findings. The main carotid plaque vulnerability characteristic assessed was the presence of a carotid ulcer. CTA and SS MRA were proven equivalent in ulcer detection, while FP MRA proved to be inferior, without, however, a statistically significant difference between modalities (*p* > 0.05) [15]. 

Trelles et al. examined the association between complex type VI carotid plaques, (based on the American Heart Association classification) and CTA findings. Type VI plaques were associated with higher maximum wall thickness [5.3 ± 0.4 mm vs. 3.9 ± 0.3 mm (*p* < 0.001)], higher maximum soft-plaque thickness [4.7 ± 0.5 mm vs. 2.2 ± 0.3 mm (*p* < 0.001)] and higher mean NASCET stenosis percentage [46.3% ± 10.6% vs. 20.5% ± 6.0% (*p* < 0.001)]. Plaques with a soft plaque thickness of less than 2.2 mm presented minimal probability of bearing vulnerable carotid plaque characteristics (Negative Predictive Value: 1) [16]. 

Gupta et al. examined the intraplaque high-intensity signal (IHIS) as a characteristic of IPH in MRA studies, and its association with CTA findings. Specifically, mean soft-plaque thickness in CTA was significantly higher in plaques with versus without IHIS presence (4.47 vs. 2.3 mm, *p* < 0.0001). In contrast, mean hard-plaque thickness in CTA was greater in plaques without versus with IHIS presence (2.09 vs. 1.16 mL, *p* = 0.0134) [8]. 

Eisenmenger et al. incorporated a magnetization-prepared rapid gradient-echo (MPRAGE) protocol for diagnosing IPH and investigated its association with adventitial calcification and an internal soft plaque (rim-sign), adventitial pattern, stenosis, maximum plaque thickness, ulceration and intraluminal thrombus on CTA. Specifically, IPH-positive plaques in MRA studies were characterized with stenosis with a higher mean NASCET percentage [53.9% vs. 24.9% (*p* < 0.001)], higher mean maximum-plaque thickness [5.93 mm vs. 3.42 mm (*p* < 0.001)], higher mean soft-plaque thickness [5.26 mm vs. 2.99 mm (*p* < 0.001)] and higher mean hard-plaque thickness [2.97 vs. 1.91 mm (*p* = 0.002)]. Based on the aforementioned CTA characteristics, IPH observed in MRA studies was mainly associated with the presence of the rim-sign in addition to increased soft-plaque thickness. This specific pattern showed excellent IPH prediction (area-under-the-curve: 0.94) [17]. 

MRA and associated CTA vulnerable plaque characteristics are reported on Table 4.

### 3.2. Risk of Bias

Time intervals between the application of the two imaging modalities were comparable, and thus, introduced a low risk of bias in that category. Moreover, the disclosure of the number and experience of researchers assessing the corresponding imaging studies of each patient introduced a low risk of bias. However, the incorporated studies were characterized by a high risk of bias, mainly regarding the absence of randomization in the patient selection process. Additionally, a not-systematic method of lesion description and match between the two imaging modalities also introduces a certain degree of bias, as solely the description of the degree of lumen stenosis was common among the included studies, incorporating the NASCET criteria. Finally, non-disclosure of some quintessential patients’ characteristics, such as age, sex and symptomatic status and cerebrovascular event severity of included patients (stroke, TIA, amaurosis fugax), negatively affects the objectivity of the produced results (Table 5, Figure 2).

## 4. Discussion

CTA and MRA have been the cornerstones, alongside with DUS for the imaging of carotid system lesions responsible for cerebrovascular events. DSA has been largely sidetracked in the last decades due to its interventional profile, related to complications. CTA has proven to be an excellent tool for stenosis evaluation, while MRA provides exquisite details regarding the morphology of the atherosclerotic carotid lesion [18,19]. Over the last decades, attention has shifted towards plaque characterization besides carotid lumen stenosis [18,19,20]. Intraplaque hemorrhage (IPH), plaque ulceration, plaque neovascularity, a thin fibrous cap and the presence of a lipid-rich necrotic core (LRNC), mainly characterized in MRA studies, have been vastly associated with cerebrovascular events, even in the absence of >50% carotid luminal stenosis [20]. Observation of those characteristics in CTA studies would greatly reduce the need for further imaging assessment, as well as allow for special patient categories (e.g., patients with cardiac pacemakers) for a complete diagnostic work-up However, stratification tools for vulnerable carotid plaques diagnosis in CTA studies have yet to be developed, as the literature lacks in comparative studies with systematic reporting of outcomes between the two imaging modalities. Studies comparing detection of vulnerable carotid plaque characteristics are few, while no systematic reviews are currently available in the literature incorporating the limited available data coherently. Thus, we opted towards the incorporation of available data, aiming towards a more coherent report.

Vulnerable carotid plaque hallmarks are more accurately distinguished in MRA studies, while some data are available regarding the appearance of these hallmarks in CTA imaging sequences. The current comparative studies suggest that certain CTA findings, including increased soft-plaque thickness, increased total-plaque thickness, increased density as well as increased NASCET percentage stenosis can be associated with vulnerable plaque characteristics detected in MRA studies. However, no current consensus exists due to the paucity of published studies as well as due to the heterogeneity of the published results.

Plaque neovascularization and IPH have been studied as complex lesion features, predisposing to acute ischemic events [21]. Increased neovascularization is the predecessor in the pathogenesis cascade of IPH, often leading to lesions related to high morbidity and mortality [22]. MRA has proven to bear excellent tissue distinguishing properties, especially in cases of atherosclerotic plaque evaluation, with high diagnostic accuracy of IPH [23,24]. Furthermore, MRA analysis of carotid plaques characterized by IPH have been associated with increased plaque progression, further destabilizing the plaque [25]. In the current review, IPH was associated with increased luminal stenosis as well as increased plaque attenuation and soft-plaque thickness [8,9,16,17]. However, non-comparative studies support that IPH could be associated with lower attenuation measurements in CTA studies, providing controversial data [26]. Thus, currently no standard CTA characteristics can be safely associated with IPH presence, as detected in MRA studies.

LRNC poses another hallmark for vulnerable carotid plaque characterization and has been evaluated in MRA studies of symptomatic and asymptomatic patients. Data suggest that LRNC findings in MRA can predict plaque progression and rupture [27,28,29]. Its clinical significance relates to treatment individualization, as aggressive medical management of plasma lipid levels in these patients could prove beneficial [29]. While none of the studies included in the current analysis evaluated LRNC, literature on comparative CTA and histology findings suggests that mainly large LRNC can be accurately observed and diagnosed with CTA. The main factor seems to be the overlap in Hounsfield densities for connective tissues and lipids, rendering it difficult to distinguish small lipid cores [30].

Ulceration, predominantly observed in DUS studies, can also be observed in MRA studies. Plaque ulcers, representing a major plaque surface anomaly, are highly related to embolic cerebrovascular events [31]. MRA studies may successfully detect carotid plaque ulcers by utilizing contrast-enhanced modalities; however, detection is depended to ulcer orientation as well as degree of lumen stenosis [32]. CTA studies have associated plaque ulcers with increased lipid-volume, increased stenosis degree and decreased calcification proportions [33,34]. Additionally, CTA studies suggest that carotid plaque ulcers generally involve extension of contrast material beyond the vascular lumen of the plaque, usually of at least 1 mm [35].

The density and thickness of the atherosclerotic plaque fibrous cap stratifies the risk of plaque rupture, as thin fibrous caps (TFC), usually overlying a necrotic core substituent, are often associated with higher rates of plaque rupture and embolic events [36]. The pathogenesis of thin fibrous cap rupture involves increased inflammation and lipid-core growth [37]. Currently, no threshold for fibrous cap thickness has been universally adopted for characterizing it as “thin”. MRA studies suggest that TFC can be accurately detected in multi-sequence imaging studies, as the observation and distinguishment of fine structures is one of the main characteristics of magnetic resonance imaging [38,39]. None of the studies included in the current review evaluated thin fibrous cap presence in MRA or CTA studies. Detection of TFC in CTA studies has proven to be difficult, as current technologies do not provide detailed enough imaging data to differentiate a thin fibrous cap from adjacent tissues.

Data from the available included studies did not support the association of IPH and ulceration detected in MRA studies with important, well-known vulnerable carotid plaque hallmarks, including thin-fibrous cap as well as lipid-rich necrotic core. Future studies could potentially examine any possible correlation between the abovementioned characteristics, which would further provide grounds for better detection of vulnerable carotid plaques in CTA studies.

### Limitations

The current descriptive systematic review bears a few limitations, mainly owed to the methodology and structure of the incorporated studies. Most importantly, there are currently no robust head-to-head comparative studies regarding vulnerable carotid plaque detection via CTA and MRA protocols. Furthermore, the retrospective nature of the included studies ascribes a certain degree of bias. In addition, the lack of systematic outcome comparison regarding common carotid plaque vulnerability characteristics, as well as the lack of symptomatic status divulgence limits the accordance of the outcomes. As knowledge on carotid plaque vulnerability rapidly expands, in parallel with diagnostic tools technical capabilities, the incorporation of patient outcomes from almost two decades ago could influence the produced results in a confusing manner. 

## 5. Conclusions

While vulnerable carotid plaque characteristics are more accurately depicted in MRA studies, CTA provides promising potential in detecting certain vulnerable lesions in risk of embolic events. Future comparative studies are essential in order to standardize the diagnostic accuracy of these two imaging modalities, as well as the association among their findings.

## Figures and Tables

**Figure 1 diagnostics-13-00646-f001:**
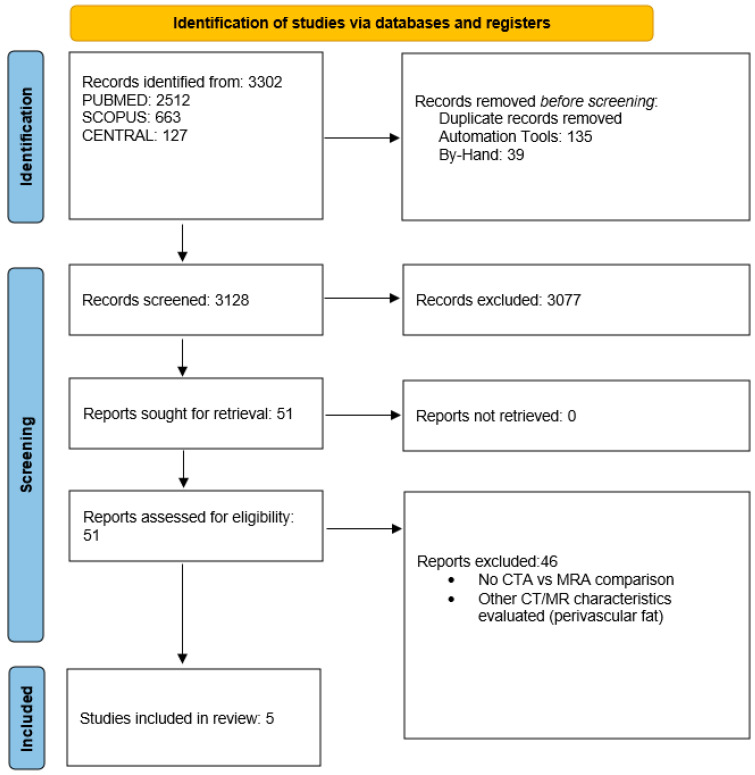
PRISMA 2020 flowchart. Screening process based on the PRISMA 2020 Guidelines for systematic reviews and meta-analyses.

**Figure 2 diagnostics-13-00646-f002:**
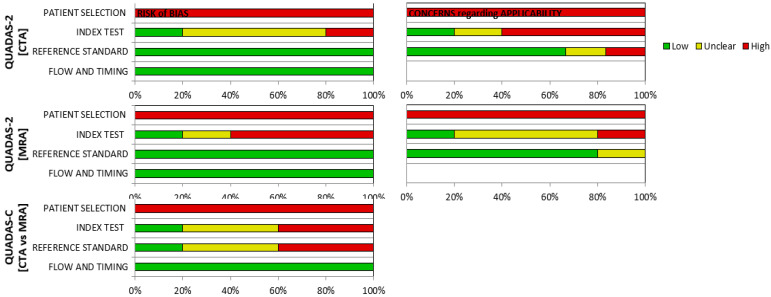
QUADAS-2 and QUADAS-C tools for diagnostic accuracy and risk of bias evaluation in imaging studies. CTA; computed tomography angiography, MRA; magnetic resonance angiography.

**Table 1 diagnostics-13-00646-t001:** Main study characteristics of the included studies.

Author	Year	Journal	Type of Study	Study Period	Study Type	Number of Patients	Age (Years)	Males	Females
U-Kind-Im et al. [9]	2010	Stroke	Retrospective of prospective data	2003–2008	Retrospective Observational	167	69 ± 12.8	109	58
Anzidei et al. [15]	2010	J Vasc Interv Radiol	Retrospective of prospective data	2008	Retrospective Observational	20	NA	NA	NA
Trelles et al. [16]	2013	Am J Neuroradiol	Retrospective of prospective data	2008–2010	Retrospective Observational	51	71.3 ± 0.9	38	13
Gupta et al. [8]	2015	Cerebrovasc Dis	Retrospective of prospective data	2009–2014	Retrospective Observational	43	74.5 ± 9.8	28	15
Eisenmenger et al. [17]	2016	Am J Neuroradiol	Retrospective of prospective data	2009–2016	Retrospective Observational	96	65.7 ± 13.4	75	21

Footnote: NA; Not available.

**Table 2 diagnostics-13-00646-t002:** Symptomatic status and stenosis severity of the included studies. Four studies incorporated data on symptomatic status and stenosis severity.

Author	Number of Patients	Number of Plaques	Symptomatic	Asymptomatic	Stroke	TIA	Stenosis Severity (NASCET) Number of Plaques (Mild/Moderate/Severe)
U-Kind-Im et al. [9]	167	319	153	14	NA	NA	193/60/66
Anzidei et al. [15]	20	40	20	0	NA	NA	7/6/17
Trelles et al. [16]	51	100	NA	NA	NA	NA	NA
Gupta et al. [8]	43	48	34	9	30	4	0/23/25
Eisenmenger et al. [17]	96	188	96	0	NA	NA	128/30/30

Footnote: TIA; transient ischemic attack, NA; Not available.

**Table 3 diagnostics-13-00646-t003:** The characteristics of the main imaging modalities (CTA, MRA), as reported in the included studies.

Author	Imaging (Control)	Imaging (Comparator)	Time Interval between Modalities (Days)	Researcher	MRA Power (Tesla)	CTA Slice (Number)	MRA Contrast—Type (Yes/No)	CTA Contrast—Type (Yes/No)
U-Kind-Im et al. [9]	MRA	CTA	21	1 neuroradiologist for MRA, 2 for CTA	1.5	4 or 64	No	Yes (Iohexol or Iodixalon)
Anzidei et al. [15]	DSA	MRA, CTA	6 ± 2	2 observers, 1 vascular radiologist	1.5	64	Yes (Gadofosveset Trisodium)	Yes (Iomeprol or Iomeron)
Trelles et al. [16]	MRA	CTA	7.8	2 radiologists	3	64	Yes (Gadobutrol)	Yes (Iopromide)
Gupta et al. [8]	CTA	MR	2	2 neuroradiologists	1.5/3	NA	No	Yes (Iohexol)
Eisenmenger et al. [17]	MRA	CTA	IPH(+): 6.9 ± 9.3, IPH(−): 5.6 ± 8.2	3 neuroradiologist (resident, fellow, attending)	1.5/3	64	Yes (NA)	Yes (Iopamidol)

Footnote: MRA; magnetic resonance angiography, CTA; computed tomography angiography, IPH; intraplaque hemorrhage.

**Table 4 diagnostics-13-00646-t004:** Association between vulnerable carotid plaque MRA and CTA characteristics as reported in the included studies.

Author	Number of Patients	Number of Plaques	Vulnerable Plaque Characteristic (MRA)	Vulnerable Plaque Characteristic (CTA)
U-Kind-Im et al. [9]	167	319	IPH (n = 56)	Mean plaque density: IPH(+): 47HU, SD: 15—IPH(−): 43HU, SD: 14 (*p* = 0.001) Mean NASCET percentage stenosis: IPH(+): 58%, SD: 26)—IPH(−): 20, SD: 26 (*p* = 0.02) Plaque ulceration: IPH(+): n = 45, n = 51 (2 readers)
Anzidei et al. [15]	20	40	Plaque ulceration (SS) (n = 9)	Plaque Ulceration (n = 8)
Trelles et al. [16]	51	100	Type VI plaque (AHA classification) (n = 23)	Maximum wall thickness: Type VI(+): 5.3 ± 0.4 mm—Type VI(−): 3.9 ± 0.3 mm (*p* < 0.001) Maximum soft plaque thickness: Type VI(+): 4.7 ± 0.5 mm—Type VI(−): 2.2 ± 0.3 mm (*p* < 0.001) Mean NASCET percentage stenosis: Type VI(+): 46.3% ± 10.6%—Type VI(−): 20.5% ± 6.0% (*p* < 0.001)
Gupta et al. [8]	43	48	IHIS (n = 27)	Mean soft-plaque thickness: IHIS(+): 4.47 ± 1.41 mm—IHIS(−): 2.32.3 ± 1.58 (*p* < 0.0001)
Eisenmenger et al. [17]	96	188	IPH (n = 44)	Mean NASCET percentage stenosis: IPH(+): 53.9%—IPH(−): 24.9% (*p* < 0.001) Mean maximum-plaque thickness: IPH(+): 5.93 mm—IPH(−): 3.42 mm (*p* < 0.001) Mean soft-plaque thickness: IPH(+): 5.26 mm—IPH(−): 2.99 mm (*p* < 0.001) Mean hard-plaque thickness: IPH(+): 2.97—IPH(−): 1.91 mm (*p* = 0.002) Rim-sign (Prevalence ratio = 11.9; 95% CI: 4.4–32, *p* < 0.001)

Footnote: MRA; magnetic resonance angiography, CTA; computed tomography angiography, SS; steady-state MRA protocol (See Text), SD; standard deviation, AHA; American Heart Association (See Text), NASCET; North American Symptomatic Carotid Endarterectomy Trial, IHIS; intraplaque high-intensity signal, IPH; intraplaque hemorrhage.

**Table 5 diagnostics-13-00646-t005:** QUADAS-2 and QUADAS-C tools. QUADAS-2 and QUADAS-C evaluation for each included study.

Study	Risk of Bias (QUADAS-2)					Applicability Concerns (QUADAS-2)				Risk of Bias (QUADAS-C)			
	P	I		R	FT	P	I		R	P	I	R	FT
		CTA	MRA				CTA	MRA					
**U-Kind-Im et al. [9]**	✗	✓	✓	✓	✓	✓	✓	✓	?	✗	✓	✓	✓
**Anzidei et al. [15]**	✗	?	✗	✓	✓	✓	✗	?	✓	✗	✗	?	✓
**Trelles et al. [16]**	✗	?	✗	✓	✓	✓	✗	?	✓	✗	?	✗	✓
**Gupta et al. [8]**	✗	?	✗	✓	✓	✓	?	✗	✓	✗	?	✗	✓
**Eisenmenger et al. [17]**	✗	✗	?	✓	✓	✓	✗	?	✓	✗	✗	?	✓

Footnote: P = patient selection; I = index test; R = reference standard; FT = flow and timing. ✓ indicates low risk; ✗ indicates high risk; ? indicates unclear risk.

## Data Availability

Appendix A provide enough information for any reader to find the data supporting our analysis.

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
