# Peer review of "Carotid Plaque Vulnerability Diagnosis by CTA versus MRA: A Systematic Review"

_diagnostics, 2023, doi:10.3390/diagnostics13040646_

Round 1

Reviewer 1 Report (Previous Reviewer 1)

I would like to thank the authors for addressing the concerns raised in the peer-review process. Though I understand their responses, I woud encourage the authors to incorporate them into the manuscript as other readers may raise the same, or similar, concerns. Their responses could be easily incorporated in the discussion.

Author Response

REVIEWER 1

  1. I would like to thank the authors for addressing the concerns raised in the peer-review process. Though I understand their responses, I would encourage the authors to incorporate them into the manuscript as other readers may raise the same, or similar, concerns. Their responses could be easily incorporated in the discussion.

Response 1: Thank you for your interesting comment. We have incorporated a number of changes in the Discussion section of our manuscript addressing the remarks of the initial reviewers.

*We have highlighted the aim of our study, reporting on the fact that while studies comparing MRA and CTA in detection of vulnerable carotid plaque characteristics are limited, no systematic review currently exists. Thus we opted towards one trying to incorporate all available data.
Page 10, Lines 251-253: Studies comparing detection of vulnerable carotid plaque characteristics are few, while no systematic reviews is currently available in the literature incorporating the limited available data. Thus we opted towards the incorporation of available data aiming towards a more coherent report.

*Based on the findings of our review, only intraplaque hemorrhage and plaque ulceration have been associated with CTA findings of vulnerable carotid plaques, while no association has been reported regarding thin-fibrous cap and lipid-rich necrotic core. We highlight this finding as well as the fact that future studies could address this problem.
Page 11, Lines 306-311: Unfortunately, data from the available included studies did not support the association of IPH and ulceration detected in MRA studies with important, well-known vulnerable carotid plaque hallmarks, including thin-fibrous cap as well as lipid-rich necrotic core. Future studies could potentially examine any possible correlation between the abovementioned characteristics which would further provide grounds for better detection of vulnerable carotid plaques in CTA studies.

Reviewer 2 Report (New Reviewer)

Only table-4 compare MRA vs CTA but incomplete MRA data, therefore comparison  between the two imaging modalities is  incomplete . Therefore the author suggested to provide valuable comments to support the aim of the study 

Author Response

  1. Only table-4 compare MRA vs CTA but incomplete MRA data, therefore comparison  between the two imaging modalities is  incomplete . Therefore the author suggested to provide valuable comments to support the aim of the study 

Response 1: Thank you for your interesting comment. All the MRA characteristics reported in Table 4 have been associated with vulnerable carotid plaques. As MRA is the main diagnostic modality used for detection of vulnerable carotid plaque characteristics, Table 4 provides which specific characteristics are observed in each study while concurrently providing details the associations between MRA and CTA vulnerable carotid plaque characteristics.

Round 2

Reviewer 2 Report (New Reviewer)

Remove the word "unfortunately".

In the limitation section ,the author must mention there are not enough to compare head to head MRA and CTA with respect to vulnerable carotid plaque in the carotid arteries 

Author Response

Reviewer 2

  1. Remove the word "unfortunately"
    Thank you for your remark. We have removed the word "unfortunately" from our discussion section. (Page 9, Line 306)

  2. In the limitation section ,the author must mention there are not enough to compare head to head MRA and CTA with respect to vulnerable carotid plaque in the carotid arteries.
    Thank you for your interesting comment. We have added this limitation as our main limitation point in the manuscript.
    (Page 9, Lines 314-316).

This manuscript is a resubmission of an earlier submission. The following is a list of the peer review reports and author responses from that submission.

Round 1

Reviewer 1 Report

The authors have performed a systematic review of the literature to evaluate the power of CTA and MRA to diagnose the vulnerabilty of carotid atherosclerotic plaques. Though the study may have some clinical value, there are some concerns that should be addressed:

1. The authors state that "increased fibrous cap (FC) thickness" is considered as vulnerable plaque feature, both in the introduction and discussion. This is a major error as it is a thin or ruptured fibrous cap what has been identified as a feature indicative of high risk for events. 

2. The goal or need for this study is not clearly stated. There is not enough information on why CTA and MRA were the imaging modalities considered in the comparison and why others, like ultrasound (which is routinely used in the clinic), were not included in the analysis. 

3. The search should have included the term "MRI". 

4. In the discussion the authors acknoledge that only the presence of hemorrhage and ulcerations were detected by both MRA and CTA, whereas other vulnerable plaque features, such as thin fibrous cap or lipid-rich necrotic core, could not been used in their comparative analysis. How can they rate the importance of these features to determine whether MRA or CTA should be used, or which provides a better diagnostic value?

5. In the conclusion, the authors state that "CTA provides promising potential in detecting certain vulnerable lesions in risk of embolic events". There was no need for a systematic review to reach this conclusion, it is already known that CTA cannot detect IPH, the status of the fibrous cap or a necrotic core. Why do the authors think that CTA has that potential?

Author Response

Point 1: The authors state that "increased fibrous cap (FC) thickness" is considered as vulnerable plaque feature, both in the introduction and discussion. This is a major error as it is a thin or ruptured fibrous cap what has been identified as a feature indicative of high risk for events. 

Response 1: Thank you for your comment. We have revised our manuscript and have corrected our previous statement.

Point 2: The goal or need for this study is not clearly stated. There is not enough information on why CTA and MRA were the imaging modalities considered in the comparison and why others, like ultrasound (which is routinely used in the clinic), were not included in the analysis. 

Response 2: Thank you for your interesting remark. While ultrasound is the first-line diagnostic modality for carotid artery imaging, we opted for CTA and MRA as we did not find any systematic reviews in the current literature of comparative studies between these two crucial diagnostic modalities, which complete the imaging work-up of patients who have undergone a cerebrovascular event.

Point 3: The search should have included the term "MRI"

Response 3: Thank you for your remark. While we certainly agree with you, unfortunately our protocol has already been registered in PROSPERO and our search analysis has been completed based on our previous search strategy. Nevertheless, we commit on performing a future analysis to examine whether the inclusion of “MRI” in our search strategy would result in different outcomes.

Point 4:  In the discussion the authors acknowledge that only the presence of hemorrhage and ulcerations were detected by both MRA and CTA, whereas other vulnerable plaque features, such as thin fibrous cap or lipid-rich necrotic core, could not been used in their comparative analysis. How can they rate the importance of these features to determine whether MRA or CTA should be used, or which provides a better diagnostic value?

Response 4: Thank you for your very interesting comment. Our analysis, based on data from the incorporated comparative studies, found significant association regarding only intraplaque hemorrhage and carotid plaque ulcerations. Unfortunately, the included studies did not examine and/or did not find any correlation between the other important vulnerable carotid artery plaque characteristics. Thus, we opted towards the discussion of these specific findings, as well as focused on the correlation provided by the current available literature.

Point 5: In the conclusion, the authors state that "CTA provides promising potential in detecting certain vulnerable lesions in risk of embolic events". There was no need for a systematic review to reach this conclusion, it is already known that CTA cannot detect IPH, the status of the fibrous cap or a necrotic core. Why do the authors think that CTA has that potential?

Response 5: Thank you for your remark. It was our intention and goal with the existing review to examine whether certain vulnerable carotid plaque characteristics detected mainly by MRA could be partly detected in CTA studies based on specific characteristics observed in the latter. Based on the available evidence, we found a possible correlation between certain MRA characteristics (intraplaque hemorrhage, ulceration) and CTA characteristics (Mean plaque density, soft and hard plaque thickness, % of stenosis based on NASCET criteria, plaque ulceration). Thus, we intended to provide data for future research where CTA studies’ characteristics could be used to predict vulnerable carotid plaque, stratifying patient follow-up and therapy.

Reviewer 2 Report

Thanks fort he opportunity to review this manuscript.

The manuscript is very interesting. The introduction is accurate and to-the-point. However, maybe in two/three lines the authors could give more insight about the imaging differences and advantages/disadvantages of CT versus MRI.

The search strategy is appropriate and follows the internationale guidelines.

The results and the discussion section are adequate.

Very minor revisions:

Table 2&3: please put the C of symptomatic and asymptomatic on the same line, same for the second chart of table 3.

Fill in DSA (digital subtraction angiography) already online 147 instead of 148

Please correct some minor grammatical mistakes in the results and discussion section

Author Response

Point 1: The manuscript is very interesting. The introduction is accurate and to-the-point. However, maybe in two/three lines the authors could give more insight about the imaging differences and advantages/disadvantages of CT versus MRI.

Response 1: Thank you for your interesting remark. We have revised our manuscript and added a short description of CTA and MRA characteristics in terms of carotid artery imaging.
Pages 4-5, Lines: 97-103: Carotid artery CTA provides a swift, detailed imaging of the extracranial and intracranial carotid artery systems, while contrast media administration aids in high accuracy detection of atherosclerotic plaques. However, specific MRA investigation protocols provide more detailed imaging due to improved spatial resolution, less saturation effects and intravoxel dephasing, and better evaluation of vessel lumen and plaque characteristics. 

Point 2: Table 2&3: please put the C of symptomatic and asymptomatic on the same line, same for the second chart of table 3.

Response 2: Thank you for your remark. We have revised the tables based on your comments.

Point 3: Fill in DSA (digital subtraction angiography) already online 147 instead of 148

Response 3: Thank you for your comment. We have revised the manuscript accordingly.

Point 4: Please correct some minor grammatical mistakes in the results and discussion section.

Response 4: Thank you for your comment. The manuscript has been revised accordingly and any observed grammatical mistake has been corrected.